# Evaluation of Medication-Related Osteonecrosis of the Jaw (MRONJ) in Terms of Staging and Treatment Strategies by Dental Students at Different Educational Levels

**DOI:** 10.3390/medicina59020252

**Published:** 2023-01-28

**Authors:** Diana Heimes, Nicolas Alexander Mark, Robert Kuchen, Andreas Pabst, Philipp Becker, Solomiya Kyyak, Daniel G. E. Thiem, Ralf Schulze, Peer W. Kämmerer

**Affiliations:** 1Department of Oral- and Maxillofacial Surgery, University Medical Center Mainz, Augustusplatz 2, 55131 Mainz, Germany; 2Private Praxis Dr. Kühnöl, Bayreuther Straße 30, 01187 Dresden, Germany; 3Institute for Medical Statistics, Epidemiology and Informatics, University Medical Center of the Johannes-Gutenberg-University Mainz, 55131 Mainz, Germany; 4Department of Oral and Maxillofacial Surgery, Federal Armed Forces Hospital, Rübenacherstraße 170, 56072 Koblenz, Germany; 5Department of Oral Surgery and Stomatology, Division of Oral Diagnostic Science, University of Bern, Freiburgstrasse 7, 3010 Bern, Switzerland

**Keywords:** panoramic radiograph, CBCT, diagnosis, MRONJ, ARONJ, dental students, qualification

## Abstract

*Background*: The role of medication-related osteonecrosis of the jaw (MRONJ) as a dento-maxillo-facial pathology is becoming increasingly important due to its growing prevalence. The success of preventive and therapeutic measures relies mainly on the dentist’s ability to correctly diagnose the disease. *Methods*: The aim of this study was to evaluate the skills of dental students of different educational levels in choosing the correct stage, diagnostics, and treatment option for MRONJ based on clinical and radiographic imaging (panoramic radiograph, CBCT). The study was designed as a cross-sectional cohort study. Twenty dental students were asked to complete a questionnaire in their third and fifth year of studies in which they had to correctly stage the disease, choose the radiological diagnostics and recommend the treatment. The control group contained experienced oral and maxillofacial surgeons. *Results*: With an overall performance of 59% (third year: 145.2/248 points; fifth year: 145.3/248 points), no statistically significant difference between the educational levels could be observed. The classification based on CBCT imaging was significantly more often correct compared to panoramic radiographs (*p* < 0.001). *Conclusions*: This study highlights students’ lack of knowledge in staging, diagnostics, and treatment of MRONJ, even though the CBCT positively affected decision-making. No significant increase in knowledge could be confirmed through clinical education. This study highlights the need for students to catch up on MRONJ diagnostics and treatment planning. Further expansion of teaching in this disease’s context and X-ray diagnostics is needed.

## 1. Introduction

Medication-related osteonecrosis of the jaw (MRONJ) is a severe debilitating condition characterized by nonhealing exposed bone in patients with a history of antiresorptive or antiangiogenic agents in the absence of radiation exposure to the head and neck region [1]. Such medications are administered to treat osteoclastic bone diseases such as osteoporosis or bone metastases in patients with solid tumors or multiple myeloma to improve bone density and arrest the development/progress of bone metastases, thereby reducing pain and the risk of pathological fractures. Bisphosphonates have a high affinity to hydroxyapatite crystals within the bone, thereby inhibiting osteoclastic resorption. Bone remodeling is further regulated by osteoblastic protein expression: receptor activator of nuclear factor-kappa-Β (RANK) ligand (RANKL) is produced by osteoblasts to promote bone resorption, whereas osteoprotegerin is a receptor which inhibits resorptive processes. Denosumab’s action mechanism, another antiresorptive agent, is the inhibition of RANKL/RANK interaction. Antiangiogenic drugs such as Bevacizumab and Sunitinub were also reported to cause MRONJ by disrupting angiogenesis-signaling cascades [1]. Three main risk factors are known to contribute to MRONJ: local risk factors, medical illness, and type of medication; the combination of those may increase the likelihood of MRONJ. It is well known that the risk of developing MRONJ is dose dependent and is accordingly determined from the nature of this underlying disease [2,3,4,5].

In 2003, Robert E. Marx was the first to suggest a relationship between non-healing exposed bone in the oral cavity and the treatment with bisphosphonates [6]. Although the first cases were reported nearly 20 years ago, the pathophysiology has not been fully understood. Hypothesized reasons for the unique localization of MRONJ exclusively in the mandible (73%) and the maxilla (22.5%) were an increased remodeling rate in the jaw, the inhibition of angiogenesis, suppression of the local immune system, a higher rate of inflammation, or infection [7].

The treatment regime is still a matter of controversy involving non-invasive (e.g., laser therapy [8], ozone [9], blood components [10]) and invasive procedures; recent data indicate beneficial outcomes in adjuvant therapy combined with surgery [11]. To date, the American Association of Oral and Maxillofacial Surgeons (AAOMS) recommends conservative treatment modalities for early stages consisting of antibiotic and antiseptic measures. Surgical approaches are indicated in patients with no treatment response or in higher stages [12,13]. In any case, early initiation of stage-specific therapy is of great importance for the prognosis of the disease. Two parameters are considered fundamental in decision making: staging and healing. The dentist’s correct classification of the disease is crucial for both [11].

Although relatively rare—the incidence of this bony disease among antiresorptive users ranges from 0.7% to 18%—the prevention and early detection of MRONJ in the dento-maxillo-facial region is of great importance due to the severe loss of life quality in affected patients [14,15,16]. Diagnostic criteria for MRONJ were developed by the AAOMS based on pharmacological history as well as clinical and radiographic features [7,13,17]. As a specialist, the dentist has an important role in preventing, early detecting, and treating those very diseases [18,19]. As a result of the presumed increasing prevalence and relevance of MRONJ in clinical practice, adequate education of dental students appears to be critical and has been reviewed in some studies to date [20,21,22,23,24,25,26]. Here, deficiencies in areas such as drug interactions and indications, risk factors for MRONJ, and preventive measures have been detected [21,22,23,24,25,26]. However, this was rarely tested using case studies reflecting the students’ ability to make decisions in the daily practice. Hence, the aim of this study was to assess the current state of knowledge of dental students in different educational levels, examine the effect of radiographic images (panoramic radiograph and cone beam computed tomography [CBCT]) on staging and treatment decision and measure the gain in knowledge during dental school using clinical case studies.

## 2. Materials and Methods

### 2.1. Study Design

The study was designed as a cross-sectional cohort study. Twenty dental students at the Johannes Gutenberg University Mainz were asked for participation in this study. If interested, a detailed explanation was given; information about the study was handed out in written form and informed consent was obtained.

### 2.2. Study Collective

In the University Medical Center of Mainz, from the very beginning of the clinical phase of study (3rd to 5th year), students are also educated in evaluating radiographic 3D data (CT and CBCT) in different lectures/courses. A first examination was performed in students in the 3rd year and a second examination in the 5th year. In the 3rd year, radiological education has been provided and the students are familiarized with the basics of the disease by attending lectures. By the 5th year, knowledge of diagnosis, classification, and treatment of MRONJ has been further deepened and tested. At this point, the students are close to graduation. The control group consisted of three experienced maxillofacial surgeons.

### 2.3. Case Studies

Patients with an MRONJ diagnosis recorded in the clinical patient management tool were screened for their suitability to participate in this study. All radiographic images included as case studies were recorded within the same year to avoid any distortion of the data due to different image acquisition quality. Of these 28 patients identified, 8 cases were finally selected, in which clinical photographs, a panoramic radiograph and a CBCT-image were available. The conditions for case selection were that they could easily be classified and represent a large diagnostic and therapeutic variability.

PowerPoint for Office 365 (Microsoft Corporation, Redmond, WA 98000, USA) was used to collect the patient data. For this purpose, the anonymous clinical intraoral images and panoramic radiographs were imported into the presentation software and the region of interest was marked. The CBCT datasets were anonymized for later review using a digital imaging and communications in medicine (DICOM) anonymizer (Rubo Medical Imaging BV, 2111XN, Aerdenhout, The Netherlands). The information on the patients’ general and special medical history (sex, age, underlying disease, duration and type of antiresorptive therapy, other risk factors and the jaw affected by the disease as well as any fistula formations of the selected patient cases) were transferred anonymously to the spreadsheet software Microsoft Excel 16 (Microsoft Corporation, Redmond, WA 98000, USA).

Clinical intraoral photographs had been taken by using a Canon EOS 100D (Canon Incorporated, Tokio, Japan). Orthophos XG Plus (Sirona Dental Systems GmbH, Bensheim, Germany) was used for 2D X-ray imaging with a current of 3–16 mA, a voltage of 60–90 kilovolts (kV) and a maximum exposure time of 14.9 s. The interpretation of these X-ray images was performed on standardized monitors using the Sidexis viewing software (Dentsply Sirona Dental Inc., York, PA, USA).

The CBCT images were obtained from two different devices: (1) 3D Accuitomo 80 (J. MORITA Corporation, Osaka 564-8650, Japan) with a voxel size of 0.08 mm, 0.125 mm, or 0.160 mm, a voltage of 60–90 kV, a current of 1–10 mA, three adjustable scan volumes (80 × 80 mm, 60 × 60 mm, 40 × 40 mm), and an exposure time of 18 s maximum. The software was One Data Viewer Plus (J. MORITA Corporation, Osaka 564-8650, Japan). (2) 3D eXam tomograph (KaVo Dental GmbH, 88400 Biberach an der Riß, Germany) with a voxel size of 0.2 mm, 0.25 mm, 0.3 mm, or 0.4 mm, a tube voltage of 120 kV, a fix tube current of 10 mA, and a maximum exposure time of 27 s. Using 3D ExamVision_software (KaVo Dental GmbH, 88400 Biberach an der Riß, Germany), the 3D DICOM datasets could be visualized. The students conducted the diagnostic evaluation of the clinical and 2D image files on a computer with the Windows 7 operating system (Microsoft Corporation, Redmond, WA 98000, USA) and an LG monitor (LG 24MB37PM-B LED with aspect ratio 16:9 and resolution: 1920 × 1080 pixels; LG Electronics Incorporated, Seoul, Republic of Korea). To read out the DICOM datasets, the subjects used 3DimViewer version 3.1.1 (3Dim Laboratory s.r.o., 62500 Brno, Czech Republic) on the Fujitsu display B22W-6 LED with aspect ratio: 16:10 and resolution 1680 × 1050 pixels (Fujitsu Ltd., Tokyo, Japan).

### 2.4. Questionnaire

A questionnaire adapted to Christoph Eisenbeiß’s questionnaire [27] for comparing radiographic imaging for the diagnosis of MRONJ was developed. This was filled out by the students on an iPad Air (Apple Incorporated, Cupertino, CA 95014, USA) via a Google form (Google LLC, Mountain View, CA 94035, USA). For the evaluation, the answers were compared with a template solution representing a consensus of three specialists in oral and maxillofacial surgery at the University Medical Center Mainz, Germany, after the completion of the study. The questionnaire consisted of ten single- or multiple-choice questions. The answers were stored anonymously online and could be retrieved for evaluation. In each case, the students first received the clinical intraoral image, then the panoramic radiograph, and finally the CBCT image. They were able to independently examine the first two images of the patient in question using a PowerPoint presentation. Due to these viewing conditions, no windowing or leveling was possible. As soon as they had completed the evaluation of the panoramic radiograph, the test supervisor opened the CBCT dataset, which the students evaluated independently on a parallel screen within the multiplanar reconstruction. Based on the clinical image, a staging of MRONJ according to AAOMS [13] had to be made first, and a distinction between stages two and three was omitted for simplification (Question 1). The definition of the AAOMS staging was given and should be applied in the following. Next, subjects were asked about their choice of treatment based on the clinical image and their choice of further radiographic diagnosis (Questions 2a and 2b). The proposed options consisted of a range of conservative and surgical treatment options based on the AWMF (Arbeitsgemeinschaft der Wissenschaftlichen Medizinischen Fachgesellschaften e.V) and AAOMS guidelines, as well as current radiographic imaging modalities. Then, regardless of the choice, the subjects received the patient’s panoramic radiograph. At this point, it was still necessary to choose whether further diagnostic means were necessary (Question 3). Specifying the region of interest, the image was graded according to radiographic features, each of which was graded from 1 (clearly visible and can be evaluated) to 2 (adequately visible but cannot be evaluated well) or 3 (not adequately visible) to 4 (not visible/very poor quality). These markers are in accordance with the literature [28,29,30,31]. The diagnostic examination of the CBCT was then performed according to the same criteria (Question 4). Finally, and after both clinical and radiographic data had been obtained, the students were again asked to classify the stage and select a therapy using the same response options as at the beginning (Questions 1a and 2a). At this point, the students were also asked to indicate which radiographic measure had the greatest influence on their choice of treatment (Question 5) (Table 1).

### 2.5. Statistics

Based on the statistical sample size calculation, *n* = 20 students were considered sufficient to answer the research question, although comparable studies had a much larger patient population [21,22,23,24,25,26]. Thus, under a level of significance of α = 0.05, with a standard deviation of 20 points and a power of 0.858, a required sample size of 14 subjects, i.e., 7 per year, was calculated.

The answers obtained using the Google form were first recorded in an Excel spreadsheet (Microsoft Corporation, Redmond, WA 98000, USA) by giving each item a specific code. This assignment was recorded on another workbook as a legend in a traceable manner. The sample solution of the three specialists was entered in the same way.

The binary data was analyzed and compared between the student and expert groups. The two groups of students were compared as unrelated samples. To test the null hypothesis that the mean scores of the two groups differed by more than 25 points, two one-sided *t*-tests (TOST) were conducted. If there was a correlation between two samples and the variables to be compared were not metric or not normally distributed, the Wilcoxon signed-rank test was used to test the hypothesis that the two samples originated from the same population. This included, for example, comparing the panoramic radiograph and CBCT of the same patient in terms of rated detectability or the choice of stage before and after diagnostic radiographic imaging. Simultaneously, differences between independent samples of the same scaling were detected by ranks using the Mann–Whitney U test. This was used, among other things, to differentiate between the 3rd and 5th year with regard to the assessment of radiographic imaging. If a Shapiro–Wilk test could not find strong evidence against normal distribution of the variables, the t-test for dependent or independent samples was used; for example, when comparing the knowledge scores of the different groups (3rd year and 5th year) regarding the choice of therapy before and after examination of the radiological diagnostics. If interrelationships of the metric variables were also to be explored in the course of the knowledge score analysis, the Pearson correlation was used for this purpose. Otherwise, the McNemar test was used to compare dependent variables with dichotomous values (f.e. comparison of the choice of the most invasive treatment and the other treatment options when considering the two time points before and after radiographic diagnosis). The Levene test was always used to detect significant differences with regard to the variance of two variables. For all tests, the null hypothesis was rejected whenever the *p* value was <0.05. In multiple testing, this significance level was adjusted to reduce α-error accumulation using Bonferroni correction. Statistical tests were performed using IBM SPSS Statistics 27 (Armonk, NY, USA).

## 3. Results

### 3.1. Clinical Staging of MRONJ According to AAOMS (Question 1)

When reducing the answers of the different educational levels of the students into binary items (agreement with gold standard/no agreement), no statistically significant difference between the groups could be found. Contrarily, it was shown that 70% of the students in both groups correctly classified the stage of disease (*p* = 1.00) (Figure 1 and Table 2).

### 3.2. Treatment Decision on the Basis of Clinical Findings (Question 2a)

Seventy-six percent (SD = 9%) of the third-year students correctly assigned patients to the treatment of choice, whereas a lower proportion of students in the third year (72 ± 15%) made the right decision (*p* = 0.5) (Figure 1 and Table 2).

### 3.3. Choice of Radiographic Imaging Technique to Be Decisive before Receiving the Radiograph (Question 2b)

A proportion of 78 ± 9% of the answers of third-year students correlated with the gold standard, whereas a lower number of students in the fifth year (75 ± 12%) chose the same radiograph to be necessary to finally decide on the treatment. Seventy-six percent of students in the third year chose the panoramic radiograph to be the necessary measure to finally decide on the treatment; the CBCT was chosen in 86.3% and the CT in only 3.8%. Students in the fifth year were more likely to choose CBCT (76.3%) over panoramic radiograph (57.5%), whereas CT has been chosen in only 3.8% of cases (Figure 1 and Table 2).

### 3.4. Evaluation Based on Panoramic Radiograph or CBCT (Question 3a and 4a)

The aim of tasks 3a and 4 of the questionnaires was to evaluate the diagnostic quality of the panoramic radiograph versus CBCT. For that purpose, students evaluated the images concerning the detectability of six radiographic characteristics (unremodeled bone and persistence of extraction sockets, dense cancellous bone, destruction of cortical bone, regions of osteosclerosis/honeycomb bone, sequester, osteolysis). The scale here ranged from 1 to 4, where 1 means “clearly visible and can be easily evaluated” and 4 indicated “not visible/very poor quality”.

The sum score of the assessability of panoramic radiographs by students at both levels of education was 2.63 ± 0.42, while the assessability of CBCT was rated statistically significantly better (2.26 ± 0.42; *p* < 0.001). Students in the third year scored the assessability of the panoramic radiograph at 2.85 ± 0.38 and the CBCT at 2.45 ± 0.28, whereas students in the fifth year rated the panoramic radiograph and the CBCT as better on average (panoramic radiograph: 2.42 ± 0.35, CBCT: 2.08 ± 0.47). Here, a statistically significant better rating of the panoramic radiograph was shown in the fifth year compared to the third year (*p* = 0.037), while this was not detectable for the CBCT (*p* = 0.082). Furthermore, the difference from the template solution in the rating of the visibility of radiographic characteristics was greatest for the CBCT (panoramic radiograph control: 2.06; CBCT control: 1.29). The answers of 30% (SD = 7%) of the third-year students and 35% (SD = 7%) of the fifth-year students (*p* = 0.167) were in accordance with the template for the evaluation of the panoramic radiograph. Evaluating the visibility of the different characteristics within the CBCT, 36 ± 16% (third year) and 42 ± 23% (fifth year) chose the answers according to the template (*p* = 0.485) (Figure 1 and Table 2). However, no conclusion can be drawn about the correctness of the answers given in each case.

### 3.5. Choice of Further Radiographic Imaging (Question 3b)

A total of 93.13% of students requested additional radiologic images after receiving the panoramic radiograph for more detailed assessment of the lesion. Among these, 89.38% preferred CBCT (third year: 96.3%; fifth year: 82.5%), 3.75% desired CT (third year: 0%; fifth year: 7.5%), and only 6.88% felt no further imaging was necessary to determine therapy (third year: 3.8%; fifth year: 10.0%). In total, 86 ± 4% of cases were answered according to the template by third-year students, whereas a lower proportion of correlation of 76 ± 20% was found between students of the fifth year and the gold standard (Figure 1 and Table 2).

### 3.6. Radiographic Staging of MRONJ According to AAOMS (Question 5a)

Before obtaining the radiological images of the patients, 30% of the cases were misdiagnosed by students in both years.

After receiving the radiological images, the percentage of misdiagnosis stayed stable with 30% in the third year and 29% in the fifth year (correct diagnosis in third year: 70 ± 18% and in the fifth year: 71 ± 23% with *p* = 0.893) (Figure 1 and Table 2).

Here, a noticeable trend toward underdiagnosis of the lesions by third-year students was evident. Eleven percent of the cases were incorrectly assigned to stage 0 and 8.8% were assigned to stage I instead of stage II/III. After receiving the radiological images, significantly fewer students chose stage 0 (5%), yet it was apparent that findings were frequently assigned to stage I rather than stages II/III, resulting in underdiagnosis here as well. In fifth-year students, misdiagnosis of stage 0 also occurred in 11.3% of cases before receiving radiological images; nevertheless, more fifth-year students chose the high stages II/III compared with third-year students (60% versus 55%). After receiving radiologic images, fewer cases were assigned to stage 0 (5%), yet the proportion choosing stage I remained the same. Here, radiologic imaging had little influence on stage selection. In the control group, which is the gold standard, stage II/III was selected more often after receiving the radiographs, leading to the higher proportion of misdiagnoses in both groups, which tended to underdiagnose the lesions.

### 3.7. Choice of Radiographic Imaging Technique to Be Decisive after Receiving the Radiograph (Question 5b)

A proportion of 85% (SD = 19%) of the students in the third year and 75% (SD = 20%) of students in the fifth year chose the imaging technique according to the gold standard. In accordance with the results on the sufficiency of panoramic radiograph for treatment planning, the vast majority of the entire student collective (80%) chose CBCT as the decisive imaging modality for evaluating the appropriate stage of the disease. Consequently, after reviewing both radiographic images, only 20% of dental students still perceived panoramic radiographs as leading the way to adequate grading. There was a preference among fifth-year students regarding the relevance of two-dimensional imaging (fifth year 25%; third year: 15%). The proportion was correspondingly lower for CBCT (fifth year: 75%; third year: 85%) (Figure 1 and Table 2).

### 3.8. Treatment Decision on the Basis of Radiographic Findings (Question 6a)

After receiving the radiological images, 66% (SD = 17%) of the third-year students and 65% (SD = 21%) of the students in the fifth year correctly assigned the patients to their respective treatment modality (*p* = 0.885) (Figure 1 and Table 2).

The data analysis of the different education levels revealed differences in the choice of treatment as well as in the consequences of the choice after receipt of the radiological imaging (Table 3). Among students at the start of their clinical education, no differences arose from the assessment of panoramic radiograph and CBCT with regard to the evaluation of a lack of need for therapy, and only minor differences arose in the choice of intravenous antibiotics administration under inpatient conditions (difference: +1.3%). In comparison to the answers of the more experienced students, a decrease in the rejection of therapy (difference: −2.5 %) and, in particular, an increased choice of systemic antibiotics (difference: +6.2 %) after completed radiological diagnostics were noted. However, both aspects did not appear significant after a comparison by the McNemar test (*p*-value “intravenous antibiosis”= 0.3; *p*-value “no therapy”= 0.5). This preference of the students of the fifth year in favor of the mentioned antibiotic application was also in correspondence with the gold standard in the before–after comparison (difference: +37.5 %). Both groups showed an almost equal tendency towards surgical treatment under general anesthesia after radiological examination (third year: −8.8% surgical treatment under local anesthesia and +8.7% surgical treatment under general anesthesia; fifth year: −7.5% surgical treatment under local anesthesia and +8.7% surgical treatment under general anesthesia). The McNemar test confirmed the statistical significance of the trend towards an invasive treatment option after receipt of the radiological images with a *p*-value of 0.038.

### 3.9. Choice of Radiographic Imaging Technique to Be Decisive for the Choice of Treatment after Receiving the Radiograph (Question 6b)

Only 21.3% of the third-year students chose the panoramic radiograph to be decisive for their choice of treatment after receiving the respective radiographs, whereas 87.5% chose the CBCT to be the most important measure to make the decision. Neither in the third nor in the fifth-year students chose the CT to be decisive after receiving the radiographs. In the fifth year, 27.5% chose the panoramic radiograph and 80% chose the CBCT to be the means of choice.

Compared to the answers before receiving the radiographs (Question 2b), the panoramic radiograph was chosen 46.2% (third year) and 30% (fifth year) less often as the imaging technique to be decisive for the choice of treatment after obtaining the respective radiographs (*p* = 0.001). The CBCT was similarly frequently selected as the means of choice before and after receiving the radiographs (third year: +1.3%; fifth year: +3.7%).

The third-year students’ responses correlated (83% (SD = 16%)) with the given responses of the specialists, whereas a lower proportion of answers (77 ± 18%) of the fifth-year students correlated with the specialists’ answers (*p* = 0.413) (Figure 1 and Table 2).

### 3.10. Evaluation of the Total Knowledge Score

To evaluate the answers given by the students, they were binarily compared with the consensus of the specialists. For each task, points could thus be scored in case of conformity with the template. No point was awarded if the respective answer did not conform; no further differentiation was made with regard to the degree of deviation from the gold standard.

Equal weighting of each possible answer resulted in a maximum score of 248 (gold standard). The two training levels were very close to each other on average, with 145.2 ± 17.55 (third year) and 145.3 ± 26.75 (fifth year) points and achieved around 59% of the maximum score. In congruence with the low mean difference of the total score, the equivalence test also showed a statistical rejection of an average difference of the scores of the educational levels by ≥25. By means of a two-one-sided *t*-test (TOST) with a *p*-value of 0.0315, the mean difference remained significantly within the equivalence limit of <25. Our null hypothesis, i.e., the exceeding of the equivalence limit of 25 in the comparison of both educational levels with regard to the total knowledge score, had to be rejected in favor of an equivalence of both training levels with regard to the mentioned aspects.

For a more detailed comparison of the correctness of questions 1 to 6b answered by students of both educational levels, weighting was performed after binary comparison to the gold standard for the purpose of equalizing original maximum score differences of the different tasks. Furthermore, after evaluating staging with the clinical image alone (Question 1), on average no divergence of the two groups could be determined at all. The largest divergence was found in Questions 3b and 5b, each with a difference of 0.1 in favor of third-year students. However, these differences were not significant after rank summation by the Mann–Whitney U test, both with regard to the question about further diagnostic means for treatment planning after evaluation of the panoramic radiograph and the inquiry about the decisive diagnostic method for the second grading decision (*p* = 0.174). Both groups (third year: mean = 0.30; fifth year: mean = 0.35) performed worst in the evaluation of radiological characteristics in the panoramic radiograph (Question 3a). In question 3a and in its equivalent in three-dimensional space (Question 4), the fifth-year students were correct more often on average than their less-experienced colleagues. However, this difference was not significant according to the Mann–Whitney U-test in both cases (question 3a: *p* = 0.101; question 4: *p* = 0.426).

## 4. Discussion

In comparison to the international literature, this study aimed not only to assess the knowledge but also to examine the applicability of what was learned in terms of the diagnosis and treatment of MRONJ. Most studies only performed knowledge assessment in a cross-sectional design and with the aid of a questionnaire. However, mainly the factual knowledge, such as the pharmacological and industrial names of the antiresorptive agents or risk factors and preventive measures regarding MRONJ, were tested [21,22,23,24].

Escobedo et al. further included constructed case studies [25,26]. In this context, students were asked to choose the appropriate procedure for the respective bisphosphonate application for different treatment options, ranging from tooth extractions to implantations to endodontic treatments. The results were then compared with a consensus of experts such as oral surgeons, including the 2009 AAOMS guideline [25,26] proposals. The present work, on the other hand, provided practical diagnostic and therapeutic assessment possibilities with clinical and radiological images of real MRONJ patients and thus stands out from comparable studies.

The questionnaire in this work extended those from other studies including postdiagnostic grading in order to detect not only the effect of radiological imaging on treatment decisions but also the change in the choice of stage, if any. Staging happened according to the AAOMS classification, and the treatment options were based on the suggestions of the German S3 guideline [7,13,32]. In combination with the complete radiological data of the patient case studies, a very realistic setting could be achieved for the students with regard to diagnostics and therapy selection, which, to the best of our knowledge, is unique in the literature to date.

The analysis of the detectability of different radiological characteristics in two- and three-dimensional images revealed a clear advantage of CBCT over panoramic radiography, as already observed by Kämmerer et al. [27,33]. In another study by Treister et al. they could show that these features were depicted more clearly in three-dimensional imaging, both in terms of quality and their extent [28]. Nevertheless, there has been no consensus in the past on the benefits of three-dimensional imaging. Chiandussi et al., for example, found no significant benefit of CT imaging in asymptomatic patients with osteonecrosis of the jaw compared with the panoramic radiograph, although they conceded that tomography provides a more accurate representation of the pathological extent overall [34]. In contrast, Stockmann et al. highlighted a greatly improved diagnostic performance of CT and MRI, while simultaneously criticizing both panoramic radiography, CT and MRI for insufficient estimation of clinical extent by these imaging modalities [35]. On the other hand, Bedogni et al. advocated the increased use of computed tomographic techniques after finding a significant correlation between histopathologic and radiologic extent with respect to the clinical extent mentioned [31]. In 2011, Cankaya et al. confirmed this correlation in an animal model also for CBCT [36]. A recent clinical study by Ristow et al. analyzed the diagnostic accuracy of panoramic radiographs and CBCT in the detection of non-vital bone changes before tooth extractions in patients with antiresorptive intake. They found the CBCT superior in sensitivity and specificity [37]. These findings are confirmed by a recent review describing panoramic radiographs to be sufficient in depicting osteolysis, osteosclerosis, and thickened lamina dura, but the CBCT and CT can show more features unique to MRONJ, such as periosteal reaction and bone-within-bone appearance. Wongratwanich et al. concluded that there is no consensus regarding the use of a specific imaging modality; therefore, dental practitioners should select the imaging modality according to the patient’s conditions to avoid over-investigation and unnecessary interventions [38].

In addition, these observations were corroborated by the evaluation of the questionnaire of the imaging decisive for the post-diagnostic stage selection and the significant preference for CBCT by the students (80%). After reviewing both diagnostics, 100% of the specialists again opted for CBCT as the decisive factor for the choice of therapy, and panoramic radiography was also selected significantly less frequently among the students (*p* < 0.001). This shift can also be traced analogously in the literature [27,33]. Similar values were provided by the results of a 2017 study by Shimamoto et al. that investigated the impact of CBCT imaging on diagnosis, treatment choice, and prognosis in stage 0 MRONJ patients. Eighty-two percent of oral surgeons reported that CBCT provided them with significant information for treatment selection, compared with 62.6% at baseline [39].

Evaluation of pre- and postdiagnostic radiographic imaging staging revealed a statistically significant difference between students’ pre- and postdiagnostic staging evaluations. Furthermore, a significant correlation between a correct evaluation of the three-dimensional imaging and the correct postdiagnostic staging choice could be shown. As in the case of staging, the evaluation of radiographic imaging also led on average to an improved assessment of the students—compared to the control group—with regard to the choice of therapy. Thus, it can be summarized that the students’ decision-making ability for adequate therapy choice improved despite relative deterioration compared to the sample solution due to the diagnostics of the radiographic imaging. Due to the significant correlation between the ability to correctly stage postdiagnostically and the subsequent correct treatment choice, it can also be cautiously assumed that the students were more capable of adequately assessing the treatment modality with the help of the imaging techniques.

The analysis of the total knowledge score summarized the already described differences of the two training levels to the sample solution and to each other. Thus, the achieved performance rate of an average of 59% of the maximum score did not appear to be extremely remarkable and, moreover, both the comparison of means and the equivalence test illustrated the absence of significant learning gains of the students in the queried content by completing the clinical semesters. Compared to the consensus-based sample solution of five experts, the evaluation by Escobedo et al. resulted in an average of 40.5% correct answers in this area. In the same question, the research group demonstrated a 13.6% improvement in students’ therapy assessments by implementing more in-depth teaching distributed among the second, fourth, and fifth years of the program [26].

This study highlights the lack of knowledge of dental students in different educational levels in evaluating MRONJ in terms of staging and need for treatment. The results of this study are in accordance with a cross-sectional questionnaire-based study by Almousa et al., who assessed dental practitioners’ and students’ knowledge of MRONJ in general. Though 68% of the 345 participants received information about antiresorptive and antiangiogenic drugs, 40% of both students and practitioners were not able to name any antiresorptive medication and a more pronounced proportion of 48% to 55% could not identify any antiangiogenic drug. Only 28% of the participants were able to name the correct definition of MRONJ according to the AAOMS. With this study, Almousa et al. were able to show that there is a great risk of missing the information of patients taking such medication, which, if not addressed, can lead to MRONJ developing in the first place and result in a significant reduction in the patient’s quality of life [40]. Those findings are in accordance with the study by de Lima et al. [23] and Franchi et al. [21]. In particular, the lack of knowledge regarding the definition of MRONJ could lead to a misdiagnosis of exposed bone and unnecessary procedures. Most of the participants were not able to identify risk factors for the development of MRONJ, which reduces the ability of the dentist to provide adequate preventive advice. These findings are further supported by the results of another cross-sectional questionnaire-based study by Al-Eid et al., who were able to show that only 35% of 74 dentists could provide a correct definition of MRONJ. Most of them did not know the medications that predispose to MRONJ [41].

The shown knowledge deficits in the prevention, diagnosis and treatment of MRONJ by both students and practitioners strongly support the urgent need for the optimization of educational programs. A high level of awareness and knowledge is needed to adequately prevent and treat such patients. The results of the studies mentioned are a reason for concern and strongly support the reinforcement of the undergraduate’s and postgraduate’s education about this pathology. Furthermore, educational programs for both dentists and physicians should be offered frequently to keep the knowledge of practitioners up to date with the current state of science and to refresh it on a regular basis. Otherwise, patients at risk could be missed, which could lead to the development of MRONJ. Furthermore, the lack of knowledge may result in a late diagnosis and unnecessary procedures, increasing the risk of more severe complications. Deficiencies in knowledge make it difficult to advise patients on the prevention of MRONJ, to diagnose them correctly and to treat them appropriately, so the dentist must currently be added as a risk factor for the development of MRONJ, whose lack of knowledge further increases the risk for the occurrence of this severely debilitating disease.

Instead of the cross-sectional design of this study, it would also have been possible to examine the same group of subjects several times as part of a longitudinal study. Since the lectures do not differ significantly between the years either in terms of content or implementation, however, no noticeable differences in the results would have been expected. Nevertheless, the chosen design resulted in the limitation that the individual progress of the test persons could not be traced. The sample size of this study with a total of 20 students was relatively small compared to similar studies with a range of 38 to 225 subjects, although a previous power analysis confirmed the sufficiency of the sample size.

The questions were asked in a closed format, mostly in single-choice format. The treatment choice and the relevant diagnostics were to be selected as multiple correct answers. This question style offered the chance for a potential bias of the results due to the correct answer selection despite insufficient background knowledge. Overall, despite the limitations, the multiple/single-choice style represents an objective and valid method to assess the quality of the results.

The aim of this study was to assess dental students of different educational levels regarding their ability to make specific oral surgical treatment choices for MRONJ based on clinical and radiographic imaging. In addition, the ability to adequately stage and the influence of radiographic assessment in the form of panoramic radiographs and datasets of CBCT on grading and treatment decision-making were reviewed. The survey was conducted to discuss a gain in knowledge in the first and last year of clinical education and was controlled with regard to its correctness by means of a comparison to the sample solution of three specialists in oral and maxillofacial surgery.

When grading six radiographic characteristics, students on average showed a significantly (*p* < 0.001) better assessment of CBCT compared to panoramic radiography. In this part, students showed their greatest weaknesses compared to other tasks in diagnosing both imaging modalities. While the students underestimated the findings during grading after obtaining clinical images and also were not adequately invasive in their choice of therapy, significant differences were shown on both sides as a result of the radiologic imaging findings in the meantime. Thus, 30% of the subjects revised their therapy decision in favor of the most invasive treatment alternative after the corresponding diagnostics.

Furthermore, with an average performance of 59% of the maximum achievable score, the mean comparison and equivalence test lacked evidence of a significant difference between the two levels of training. Nevertheless, to the best of our knowledge, the diagnostic capabilities to this extent and the transfer requirements based on them for students to choose an oral surgery treatment strategy are unique to date, and thus they complicate the detailed comparison with other studies on this.

## 5. Conclusions

This study once again highlighted the need for students to catch up on this topic, although CBCT in particular had a positive effect on the subjects’ decisions overall. Furthermore, no significant increase in knowledge could be confirmed through clinical training. Based on these results and since the implementation of an MRONJ seminar to deepen student knowledge has already achieved positive results in the past, further expansion of teaching in the context of this disease as well as (three-dimensional) X-ray diagnostics seems logical.

## Figures and Tables

**Figure 1 medicina-59-00252-f001:**
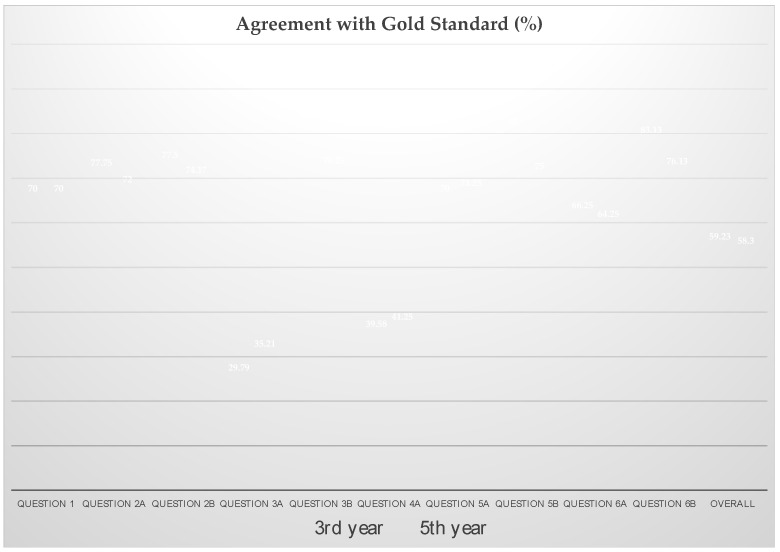
Proportion of correct answers by students of the 3rd and 5th year of studies. Question 1: Clinical staging of MRONJ according to AAOMS. Question 2a: Treatment decision on the basis of clinical findings. Question 2b: Choice of radiographic imaging technique to be decisive before receiving the radiograph. Question 3a: Evaluation based on panoramic radiograph. Question 3b: Choice of further radiographic imaging. Question 4a: Evaluation based on panoramic CBCT. Question 5a: Radiographic staging of MRONJ according to AAOMS. Question 5b: Choice of radiographic imaging technique to be decisive after receiving the radiograph. Question 6a: Treatment decision on the basis of radiographic findings. Question 6b: Choice of radiographic imaging technique to be decisive for the choice of treatment after receiving the radiograph.

**Table 1 medicina-59-00252-t001:** Questionnaire.

1.1 Staging (clinical)	
Staging according to American Association of Oral and Maxillofacial Surgeons (AAOMS)	
Stage	Definition
0	Patients with no clinical evidence of necrotic bone, but non-specific clinical findings, radiographic changes and symptoms
1	Exposed and necrotic bone, or fistulae that probes to bone, in patients who are asymptomatic and have no evidence of infection
2	Exposed and necrotic bone, or fistulae that probes to bone, associated with infection as evidenced by pain and erythema in the region of the exposed bone with or without purulent drainage
3	Exposed and necrotic bone or a fistula that probes to bone in patients with pain, infection, and one or more of the following: exposed and necrotic bone extending beyond the region of alveolar bone, (i.e., inferior border and ramus in the mandible, maxillary sinus and zygoma in the maxilla) resulting in pathologic fracture, extra-oral fistula, oral antral/oral nasal communication, or osteolysis extending to the inferior border of the mandible of sinus floor
Question 1	According to the classification of AAOMS: how would you classify the medical finding (see photo)?
	○ 0 ○ 1 ○ 2–3
**1.2. Choice of treatment (clinical)**	
Question 2a	Based on the clinical signs (see photo), which treatment would you choose (multiple-choice)?
	○ Oral antibiotics ○ Surgery under local anesthesia ○ Stationary admission for intravenous antibiotics ○ Surgery under general anesthesia ○ No therapy needed
Question 2b	Which radiological diagnostics do you consider necessary to finally decide on the treatment (multiple-choice)?
	○ Panoramic radiograph ○ CBCT ○ CT
**1.3. Panoramic radiograph**	
Question 3a	Can the following findings be detected in region 14 and if so, how well? (1 = clearly visible and can be easily evaluated; 2 = adequately visible, but cannot be easily evaluated; 3 = not adequately visible; 4 = not visible/very poor quality)
	○ Unremodeled bone and persistence of extraction sockets ○ Dense cancellous bone ○ Destruction of cortical bone ○ Regions of osteosclerosis/honeycomb bone ○ Sequester ○ Osteolysis
Question 3b	Will you request additional diagnostic resources for treatment planning?
	○ No ○ CBCT ○ CT
**1.4. CBCT**	
Question 4a	○ Can the following findings be detected in region 14 and if so, how well? (1 = clearly visible and can be easily evaluated; 2 = adequately visible, but cannot be easily evaluated; 3 = not adequately visible; 4 = not visible/very poor quality)
	○ Unremodeled bone and persistence of extraction sockets ○ Dense cancellous bone ○ Destruction of cortical bone ○ Regions of osteosclerosis/honeycomb bone ○ Sequester ○ Osteolysis
**1.5. Staging (radiological)**	
**Staging according to American Association of Oral and Maxillofacial Surgeons (AAOMS)**	
Stage	Definition
0	Patients with no clinical evidence of necrotic bone, but non-specific clinical findings, radiographic changes and symptoms
1	Exposed and necrotic bone, or fistulae that probes to bone, in patients who are asymptomatic and have no evidence of infection
2	Exposed and necrotic bone, or fistulae that probes to bone, associated with infection as evidenced by pain and erythema in the region of the exposed bone with or without purulent drainage
3	Exposed and necrotic bone or a fistula that probes to bone in patients with pain, infection, and one or more of the following: exposed and necrotic bone extending beyond the region of alveolar bone, (i.e., inferior border and ramus in the mandible, maxillary sinus and zygoma in the maxilla) resulting in pathologic fracture, extra-oral fistula, oral antral/oral nasal communication, or osteolysis extending to the inferior border of the mandible of sinus floor
Question 5a	According to the classification of AAOMS: how would you classify the medical finding after obtaining the radiological diagnostics?
	○ 0 ○ 1 ○ 2–3
Question 5b	Which diagnostic (panoramic radiograph or CBCT) was decisive?
	○ Panoramic radiograph ○ CBCT
**1.6. Choice of treatment (radiological)**	
Question 6a	Based on the radiological signs which treatment would you choose (multiple-choice)?
	○ Oral antibiotics ○ Surgery under local anesthesia ○ Stationary admission for intravenous antibiotics ○ Surgery under general anesthesia ○ No therapy needed
Question 6b	Which diagnostic (panoramic radiograph or CBCT) was decisive?
	○ Panoramic radiograph ○ CBCT

**Table 2 medicina-59-00252-t002:** Proportion of correct answers and agreement with the gold standard.

	**Correct Answers/Agreement with Gold Standard (%)**	
**Question**	3rd grade	5th grade
**Clinical staging of MRONJ according to AAOMS (Question 1)**	70	70
**Multiple choice:** **Treatment decision on the basis of clinical findings (Question 2a)**		
**Answer 1**	75	65
**Answer 2**	63.75	61.25
**Answer 3**	70	71.25
**Answer 4**	71.25	66.25
**Answer 5**	98.75	96.25
**Multiple choice:** **Choice of radiographic imaging technique (Question 2b)**		
**Answer 1**	67.5	57.5
**Answer 2**	68.75	68.75
**Answer 3**	96.25	96.25
**Multiple choice:** **Evaluation based on panoramic radiograph (Question 3a)**		
**Answer 1**	46.25	48.75
**Answer 2**	22.5	36.25
**Answer 3**	43.75	35
**Answer 4**	12.5	25
**Answer 5**	28.75	40
**Answer 6**	25	26.25
**Choice of further radiographic imaging (Question 3b)**	86.25	76.25
**Multiple choice:** **Evaluation based on CBCT (Question 4a)**		
**Answer 1**	43.75	46.25
**Answer 2**	31.25	25
**Answer 3**	68.75	52.5
**Answer 4**	18.75	43.75
**Answer 5**	30	15
**Answer 6**	45	65
**Clinical staging of MRONJ after radiographic imaging according to AAOMS (Question 5a)**	70	71.25
**Decisive diagnostic (panoramic radiograph or CBCT) (Question 5b)**	85	75
**Multiple choice:** **Treatment decision on the basis of radiographic findings (Question 6a)**		
**Answer 1**	68.75	58.75
**Answer 2**	65	61.25
**Answer 3**	43.75	52.5
**Answer 4**	55	50
**Answer 5**	98.75	98.75
**Multiple choice:** **Decisive diagnostic (panoramic radiograph or CBCT) (Question 6b)**		
**Answer 1**	78.75	72.25
**Answer 2**	87.5	80
**Overall agreement**	59.23	58.3

**Table 3 medicina-59-00252-t003:** Choice of treatment before and after receipt of radiological images.

		Time Point in Relation to Receipt of Radiological Imaging		
Year	Treatment modality	Before (%)	After (%)	Difference (%)
3rd	Oral antibiotics	35	31.3	−3.7
	Surgery under local anesthesia	43.8	35	−8.8
	Intravenous antibiotics	42.5	43.8	+1.3
	Surgery under general anesthesia	46.3	55	+8.7
	No treatment	1.3	1.3	0
5th	Oral antibiotics	40	41.3	+1.3
	Surgery under local anesthesia	46.3	38.8	−7.5
	Intravenous antibiotics	46.3	52.5	+6.2
	Surgery under general anesthesia	41.3	50	+8.7
	No treatment	3.8	1.3	−2.5

## Data Availability

Data is contained within the article. The raw data are available from the corresponding author on reasonable request.

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
