# Peer review of "Evaluation of Medication-Related Osteonecrosis of the Jaw (MRONJ) in Terms of Staging and Treatment Strategies by Dental Students at Different Educational Levels"

_medicina, 2023, doi:10.3390/medicina59020252_

Round 1
Reviewer 1 Report
Dear Authors,
The study is very well conducted and designed, the questionnaire and the methods of administration of the same are valid and the statistics are well conducted. However I wonder what is the value of a study that tests the knowledge of a subject by a limited number of students all belonging to the same university. As if that weren't enough, the authors underline the inadequacy of the preparation of the 20 students involved, more than a study it seems to me an admission of guilt given that it is assumed that the students were also partly trained by the authors of this manuscript.
I don't think this article should be rejected, but I believe that this study needs to be expanded in the number of students involved and that it should involve more dental schools, and for this reason I believe that, for the moment, it should be rejected.
Best regards
Author Response
We thank the reviewer for their efforts. This study should be seen as a pilot study followed by further, extended analyses of students' current knowledge regarding various highly relevant topics in Oral and Maxillofacial Surgery. Especially regarding the change of the dental licensing regulations in Germany in 2022, the question of the difference between students in the old and the new licensing rules arises. Since the education of students is central to the care of patients and should be subjected to constant critical (also self-critical) analysis, we see in the conducted study, especially given the time-consuming analysis of several clinical cases by an - admittedly - small number of students, a high value for the improvement of teaching in this field. As the review does not raise any constructive points about the present article, we hope to have satisfied review #1 with this statement.
Reviewer 2 Report
The manuscript “Evaluation of Medication related Osteonecrosis of the Jaw (MRONJ) in terms of staging and need for treatment by dental students at different educational levels” is a cross-sectional cohort study whose aim is “to assess the current state of knowledge of dental students in different educational levels, examine the effect of radiographic images (panoramic radiograph and cone beam computed tomography [CBCT]) on staging and treatment decision and measure the gain in knowledge during dental school using clinical case studies”.
The paper falls within the scope of the journal.
English language and style are fine/minor spell check is required.
In my opinion, the paper possesses some flaws, in detail:
- In the title, it is a little bit complex to read, please try to revise it. Probably, “need for treatment” could be “treatments’ strategies”;
- Abstract, I think that the number could be avoided in the structured abstract;
- Introduction, the authors stated that “…osteoporosis or bone metastases with the aim of improving bone density, thereby reducing pain and the risk of pathological fracture”. In my opinion, it is important to define that bone metastases are in patients affected by solid tumors, that also patients affected by multiple myeloma are included, and that the medications also arrest the development/progress of bone metastases.
- Materials and Methods, the ethics committee approval number is not reported;
It is not clear why the authors omitted for simplification a distinction between stages 2 and 3 (page 3, lines 144-145). In my opinion, stages 2 and 3 are very different; moreover, if “Based on the clinical image”, it is more difficult to distinguish stage 1 from stage 2 (i.e., the only difference is the presence of symptoms in the AAOMS staging system).
Why did the authors decide to test only the capability of students to evaluate only the staging and need for treatment and to avoid the diagnosis?
Additionally, are the AWMF guidelines only in German, or are available also in English?
- References, n.7 and n.16 are the same. Regarding the role of CBCT, there are several more recent papers available.
It is very interesting that this study highlight that “students on average showed a significantly (p<0.001) 435 better assessment of CBCT compared to panoramic radiography” because I strongly believe that CT/CBCT should be requested in any case of suspected MRONJ. However, overall, I have concerns regarding the contribution of the paper to the literature, also due to the small sample size.
Author Response
We thank the reviewer for their efforts and constructive comments.
Comment #1: In the title, it is a little bit complex to read, please try to revise it. Probably, “need for treatment” could be “treatments’ strategies”;
Answer: Thank you for this note. We changed the title according to the suggestion
Comment #2: Abstract, I think that the number could be avoided in the structured abstract;
Answer: We deleted the numbers according to the reviewer’s suggestion.
Comment #3: Introduction, the authors stated that “…osteoporosis or bone metastases with the aim of improving bone density, thereby reducing pain and the risk of pathological fracture”. In my opinion, it is important to define that bone metastases are in patients affected by solid tumors, that also patients affected by multiple myeloma are included, and that the medications also arrest the development/progress of bone metastases.
Answer: We added this information (Lines 44, 45)
Comment #4: Materials and Methods, the ethics committee approval number is not reported;
Answer: The study was conducted according to the guidelines of the Declaration of Helsinki. In the federal state of Rhineland-Palatinate anonymized, retrospective data evaluations do not have to be reported to the local ethics committee.
Comment #5: It is not clear why the authors omitted for simplification a distinction between stages 2 and 3 (page 3, lines 144-145). In my opinion, stages 2 and 3 are very different; moreover, if “Based on the clinical image”, it is more difficult to distinguish stage 1 from stage 2 (i.e., the only difference is the presence of symptoms in the AAOMS staging system).
Answer: Since, in this survey, oral clinical pictures were particularly used to diagnose the disease, the decision was made to reduce the answers to stages 0, 1, and 2-3. This decision is derived from the typical characteristics of each stage. Stage 0 is characterized by the absence of any clinical signs of MRONJ and, in particular, by intact mucosa. In stage 1, bone is exposed; this stage differs from stage 2 by the absence of other clinical symptoms - especially the absence of infection. Signs of infection, i.e., pus or erythema, are present in stage 2. This stage differs from stage 3 only by the extent and the presence of complications. We found it difficult to make this distinction on the basis of a clinical, enoral picture, which is why we decided to merge the two stages. Nevertheless, it is understandable that there may be different opinions based on a different reasoning structures. In this respect, the proposed classification suggested by Reviewer 2 would also be acceptable. However, we do not think that the quality of the study is diminished by the simplification of this question. In this respect, we hope for the editor's understanding of the choice of questions in this study.
Comment #6: Why did the authors decide to test only the capability of students to evaluate only the staging and need for treatment and to avoid the diagnosis?
Answer: We decided to use the present questionnaire structure and not to use differential diagnosis because we think that, in some cases, it is difficult to differentiate between MRONJ and other similar disease patterns on the basis of a clinical and radiological image alone. We decided to use the present questionnaire structure and not to use differential diagnosis because we think that, in some cases, it is difficult to differentiate between MRONJ and other similar disease patterns on the basis of a clinical and radiological image alone. For example, exposed bone can also result from a pressure site alone, which is clinically indistinguishable from bisphosphonate-associated necrosis without the patient's history. Tumor diseases (multiple myeloma) can also lead to healing delays after tooth extractions or exposed bone due to denture pressure points - the radiological picture cannot always be clearly distinguished from MRONJ. Due to a large amount of possible differential diagnoses, the difficulty of asking this in a structured way in a questionnaire, and the clear radiological-imaging character of the study, we decided to use the present structure. We hope for your understanding in this regard.
Comment #7: Additionally, are the AWMF guidelines only in German, or are available also in English?
Answer: To my knowledge, the AWMF guidelines are only available in German, but I added the AAOMS position paper as a reference.
Comment #8: References n. seven and n.16 are the same. Regarding the role of CBCT, there are several more recent papers available.
Answer: We corrected this mistake and added more recent articles as references.
Comment #9: Please find a corresponding publication to the German AWMF guideline. AAOMS MRONJ guideline?
Answer: Please refer to Answer #7.
The question of whether panoramic radiographs, CBCT, CT, or other imaging modalities should be preferred in diagnosing MRONJ is now extensively addressed in lines 342-362.
Kind regards
Round 2
Reviewer 1 Report
My doubts remain and, given the opinion of the other reviewer, having no comments regarding method corrections, I approve the publication despite my initial opinion.